# Improved Isolation of Ultra-High-Molecular-Weight Genomic DNA Suitable for Third-Generation Sequencing

**DOI:** 10.3390/microorganisms13030534

**Published:** 2025-02-27

**Authors:** Ayşe Öykü Ova, Enrique Joffre, Reza Zandi Shafagh, Mariana F. G. Assunção, Roman Y. Sidorov, Lilia M. A. Santos, Volker M. Lauschke, Ute Römling

**Affiliations:** 1Department of Microbiology, Tumor and Cell Biology, Biomedicum, Karolinska Institutet, 17177 Stockholm, Sweden; 2Faculty of Science, Department of Molecular Biology and Genetics, Izmir Institute of Technology, Urla, 35430 Izmir, Turkey; 3Department of Physiology and Pharmacology, Karolinska Institutet and Center for Molecular Medicine, Karolinska University Hospital, 17177 Stockholm, Swedenvolker.lauschke@ki.se (V.M.L.); 4Dr. Margarete Fischer-Bosch Institute of Clinical Pharmacology, 72076 Stuttgart, Germany; 5University of Tübingen, 72074 Tübingen, Germany; 6Division of Micro- and Nanosystems, KTH Royal Institute of Technology, 11428 Stockholm, Sweden; 7Department of Life Sciences, Coimbra Collection of Algae (ACOI), University of Coimbra, Calçada Martim De Freitas, 3000-456 Coimbra, Portugal; 8Department of Pharmacy, The Second Xiangya Hospital, Central South University, Changsha 410017, China

**Keywords:** bacteria, DNA isolation, incubation chamber, third-generation sequencing

## Abstract

Although a variety of protocols to isolate high-molecular-weight genomic DNA exist, the isolation and preservation of ultra-high-molecular-weight genomic DNA of sufficient quality and length for error-free third-generation sequencing remains challenging. Inspired by the isolation of high-molecular-weight DNA in agarose plugs suitable to be separated by pulsed-field gel electrophoresis, we report the construction of an incubation chamber for DNA isolation. The chamber is flanked by semi-permeable polycarbonate membranes of variable pore size for the selective diffusion of compounds and components, which allows efficient cell lysis and the subsequent isolation of ultra-high-molecular-weight genomic DNA without shearing. The designed experimental approach is simple, but effective in isolating high-quality ultra-long genomic DNA that is suitable for third-generation sequencing by Oxford Nanopore Technology from challenging bacterial samples. We envisage that genomic DNA isolation using the incubation chamber, or variations thereof, will facilitate the regular sequencing of megabasepair-long DNA fragments, with a multitude of applications in microbiology, immunology, oncology, paleontology and forensic science.

## 1. Introduction

DNA is the central macromolecule in any living cell for the storage and transmission of genetic information to offspring. Alterations, such as recombination events in the DNA sequence, occur inherently during development, but other rearrangements of the genomic information can lead to inherited diseases that can be hard to trace due to being located in repetitive regions and/or introns [1,2]. The high chemical stability of DNA enables tracing life throughout earth history dating back up to millions of years [3]; and considering DNA for the long-term storage of information [4]. Recognized in 1869 by Friedrich Miescher as a novel macromolecule [5], its double-helical structure was eventually identified in 1953 [6,7].

The established protocols for the isolation of genomic DNA proceed via several major steps, with the efficient breaking up of cells and the removal of chemically different macromolecules and small molecular contaminants being essential parts. The disruption of cells for high-molecular-weight genomic DNA isolation can be a major challenge depending on the isolation source, and may include the physical (including mechanical or heat), enzymatic, or chemical lysis of cells. In the case of bacteria, the stabilizing cell wall, in particular the peptidoglycan sacculus, can be degraded by treatment with the peptidoglycan *N*-acetylmuramoylhydrolase lysozyme, or alternative peptidoglycan-degrading enzymes, while the stabilizing cell walls of fungi can be degraded via digestion with zymolyase, comprising a mixture of cell-wall-degrading enzymes with various substrate specificities [8]. A broad variety of protocols are available for the downstream isolation of genomic DNA from organisms throughout the phylogenetic tree. To effectively remove small molecules and other common macromolecules besides DNA without damaging the DNA structure and composition, the most widely used protocols rely on the enzymatic removal of RNA via bovine pancreas isolated RNase A, the digestion of proteins by proteinase K, and the solubilization of lipid-bilayer membranes by detergents such as sodium dodecylsulfate or N-lauroylsarcosine. Proteins can also be effectively denatured with chaotropic agents such as guanidinium chloride prior to removal. Residual proteinaceous components, lipids and other components can subsequently be effectively removed by extraction with organic solvents such as CHCl_3_ and phenol. Alternatively, DNA can be reversibly bound to a silica matrix column loaded with organic or inorganic oppositely charged molecules through electrostatic interactions, or to matrix-coated paramagnetic beads or nanodisks, which also allow the size fractionation of DNA.

In addition to these basic steps of DNA isolation, depending on the organism, additional extraction steps can be introduced that are adapted to the metabolic and morphological complexity of the organism, such as the presence of an extensive amount of (exo)polysaccharides or secondary metabolites, such as polyphenolic compounds that inhibit the downstream enzymatic manipulation of the isolated DNA. Such isolation steps include the use of N-cetyl-N,N,N-trimethylammonium bromide (CTAB), a quaternary ammonium salt detergent, to precipitate acidic polysaccharides and polyvinylpyrrolidone for the scavenging of polyphenols [9]. Alternative approaches use the enzymatic digestion of (exo)polysaccharides [10].

Whether a simple or more complex isolation protocol is applied, all protocols consist of several consecutive steps for genomic DNA isolation, which increases the likelihood of shearing the DNA by pipetting, vortexing or centrifugation. We aimed to develop a DNA isolation approach that consists of as few manipulative steps as possible to gain highly pure unsheared genomic DNA. To this end, we adapted the previously developed DNA isolation procedure for the preparation of ultra-high-molecular-weight DNA (>1 Mbp) for pulsed-field gel electrophoresis [11,12,13] in order to isolate highly pure, ultra-high-molecular-weight DNA from Gram-negative and Gram-positive bacteria for downstream processing. To this end, we designed and constructed an incubation chamber that can hold variable volumes of cell suspension, with a default of 200 µL, confined by semi-permeable polycarbonate membranes with a 0.4 µm pore diameter. The incubation of the chamber content in various buffers—lysozyme buffer and subsequently a high-pH lysis buffer containing a chelating agent, a detergent, and proteinase K—yielded high-quality high-molecular-weight genomic DNA suitable for downstream third-generation sequencing. Our approach is reliable and resource- and labor-saving, and is thus suitable to replace more complex DNA isolation protocols from different origins, with the potential to subsequently obtain ultra-long sequence reads.

## 2. Materials and Methods

### 2.1. Fabrication of the Incubation Chamber

The incubation chamber consisted of three polycarbonate parts, with dimensions of the different pieces as outlined in Figure 1. The fabrication of the incubator parts was performed by precision micromilling of polycarbonate (PC) sheets using a MiniMill GX (Minitech Machinery Corp., Norcross, GA, USA). The parts were assembled through six threaded holes on the central part, along with counterpart non-threaded holes in the sandwiching parts (plugs), with a close fit clearance for M3 polypropylene (PP) screws. The incubation chamber was constructed to be flanked on both sites by replaceable, tightly fixed semi-permeable polycarbonate membranes with a default pore size of 0.4 µm. Semi-permeable membranes with alternative pore size can also be used. The dimensions of the chamber were adjusted to fit the device into a 50 mL centrifuge tube for incubation. The authors can be contacted for further detailed information.

### 2.2. Microbial Strains and Growth Conditions

The microbial strains used to test the DNA isolation protocols in the incubation chamber were *Agrobacterium* sp. AM large and small variants, *Pseudoarthrobacter* sp. M63 and *Arthrobacter* sp. CM63 isolated from *Micrasterias papillifera* ACOI 247 algal culture (ACOI [14]), two saprophytic *Pseudomonas* sp. isolates (PRS9 and GRP3) [15], *Pseudomonas aeruginosa* UMR1 ear isolate (unpublished) and *P. aeruginosa* SG17M, a mucoid and biofilm (pdar28) variant of *Klebsiella pneumoniae* GT1 (DSM 12082), *Escherichia coli* Fec59 and Fec101 [16], a Gram-positive *Enterococcus muntii* HF strain isolated from horse feces, and *Staphylococcus* sp. W15 isolated from water (this work).

Strains were stored at −80 °C and were recovered by growth on a LB (Luria–Bertani) or blood agar plate (*E. muntii* HR and *Staphylococcus* sp. W15) overnight at either 20 or 37 °C. Prior to DNA isolation, strains were incubated in liquid culture (LB medium or LB with 5% horse blood (*E*. *muntii* HR, *Staphylococcus* sp. W15)) until the logarithmic phase of growth (OD_600_ approx. 1), or on a LB agar plate overnight at room temperature (*Agrobacterium* sp. AM small, *Pseudoarthrobacter* sp. M63).

### 2.3. Identification of Bacterial Isolates by 16S rDNA Sequencing

Bacterial isolates of previously unknown phylogeny recovered from algal culture, horse feces and water were identified by 16S rDNA sequencing. A PCR product of 16S rDNA was amplified using the standard primers 27F and 1492R, and subsequently Sanger sequenced using the Eurofins Mix2SeqKit (Eurofins Genomics Europe, Ebersberg, Germany), and its sequencing service. 16S rDNA sequences were blasted against the NCBI database for genus/species identification [17].

### 2.4. Digestion of Genomic DNA in the Incubation Chamber

The bacterial cell suspension was adjusted to OD = 10 as the default parameter. After the inclusion of the bacterial cell suspension in the incubation chamber, covered by 0.4 µm of polycarbonate membrane (Isopore^TM^ (Merck Millipore, Darmstadt, Germany)), the entire chamber was incubated in 10 mL of buffer in a 50 mL Eppendorf incubation tube. Incubation took place solely with lysis buffer (0.5 mg/mL proteinase K in 0.5 M EDTA (ethylenediaminetetraacetic acid), 100 mM NaCl, 1% N-lauroylsarcosine, pH 9.5) for 24 h at 56 °C (core digestion procedure). A high pH and detergent provide optimal conditions for the catalytic activity of the serine protease proteinase K, with EDTA to inhibit Mg^2+^-dependent nucleases. Optionally, the incubation could be carried out with lysozyme buffer (10 mg/mL lysozyme in 100 mM NaCl, 20 mM Tris-HCl, pH 5.0) for 2–4 h at 37 °C prior to cell lysis. After incubation, the incubation chamber was rinsed several times with autoclaved ultrapure water and subsequently incubated two times for at least 8–10 h each in 50 mL of sterile H_2_O.

Upon further purification, the volume of the DNA solution was at least doubled with H_2_O (or adjusted to obtain a solution), and the contaminants were extracted with an equal volume of CHCl_3_. The solution was subsequently adjusted to 0.3 M NaAcetate and a pH of 5.2, and the genomic DNA was precipitated with 2.5 volumes of ethanol, washed with 70% ethanol, and air-dried. Subsequently, the genomic DNA was dissolved in 200 µL double-distilled autoclaved H_2_O.

### 2.5. DNA Quality Control

The concentration and the spectrum of the isolated genomic DNA were measured using a NanoDrop^TM^ 2000c spectrophotometer (Thermo Fisher Scientific, Waltham, MA, USA). The genomic DNA was subsequently run on 0.8% agarose gel in 1× TAE (Tris-Acetate-EDTA) buffer to assess integrity and RNA contamination.

### 2.6. Assessment of Length Distribution of the Genomic DNA

The assessment of the length distribution of the genomic DNA was performed with pulsed-field gel electrophoresis using a CHEF-DR III apparatus (Biorad, Hercules, CA, USA). The gel running conditions were 1.5% agarose gel; 0.5x TBE (Tris-Boric acid-EDTA) buffer; electric field strength: 6 V/cm; angle: 120°; temperature: 8 °C; and linear pulsing from 10 to 30 sec for 30 h, followed by a second linear ramp from 50 to 70 sec for 16 h. The gel was stained with 0.5 µg/mL ethidiumbromide for 45 min, followed by three times destaining in double-distilled water for 30 min each. Phage Lambda DNA (48.5 kbp) and *Saccharomyces cerevisiae* W303-translocated DNA served as size standards.

### 2.7. Nanopore Sequencing

The rapid sequencing DNA V14 barcoding (SQK-RBK114.24) kit (Oxford Nanopore Technologies, Oxford, UK) with the standard protocol was used for DNA sequencing on an R10.4.1 MinION Flow Cell on a MinION device (Oxford Nanopore Technologies, Oxford, UK), according to the manufacturer’s instructions. Sequencing was performed for 48 h. Downstream quality analysis of the fastaq sequencing data was performed using the NanoForms pipeline with standard parameters [18]. Raw sequencing reads were submitted to the SRA archive at the National Center for Biotechnology Information (NCBI) under the BioProject ID submission code PRJNA1219145.

## 3. Results

To develop a straightforward, reliable and efficient DNA isolation procedure using a substantially different approach to those described previously, we were inspired by the isolation of ultra-long megabasepair-size DNA in agarose plugs, suitable to be applied in pulsed-field gel electrophoresis [13]. In this procedure, whole cells are embedded in low-melting agarose plugs, with the DNA subsequently isolated by the incubation of the plugs in a lysis buffer containing an ion chelator, a detergent and a proteinase (for buffer composition, see Experimental Procedures) [12]. A high alkaline pH of 9.0 to 9.5 in the buffer ensures the chemical degradation of RNA. Subsequently, the lysis buffer is removed through several washing steps with Tris-EDTA buffer. A prior lysozyme digestion step is included to efficiently lyse the cell wall, which is particularly relevant for Gram-positive bacterial cells, including *Mycobacterium* spp. This described procedure is reliable and applicable (in variations) to isolate genomic DNA from all organisms, including bacteria, archaea, fungi and other eukaryotes, including humans [19,20].

In order to mimic DNA preparation in agarose plugs, we developed an incubation chamber with the principle being to lyse cells and concomitantly digest contaminants in a chamber flanked by a semi-permeable membrane without the application of pipetting or centrifugation steps.

The designed chamber consists of an annulus: a punctured disk where one inner ring is flanked by two outer rings in concentric circles (Figure 1). The inner rings embrace a central cavity dimensioned to harbor a defined incubation volume (200 or 1000 µL), flanked on each side by outer wider cavities. The central space is covered on both sides with a semi-permeable membrane, which is subsequently fixed by an annulus that tightly holds the membrane in place by covering the space between the inner and outer holes. The adaptor can be tightly connected to the central disk with six screws, and thus firmly closes the incubation chamber.

We subsequently chose different Gram-negative and Gram-positive bacteria to test the quality and yield of the DNA isolated in the incubation chamber. We first chose two *Pseudomonas* sp. strains and one *Pseudomonas aeruginosa* strain, and then harvested the cells grown in LB medium in the logarithmic phase at OD_600_ = 1. Next, we placed 200 µL of the cell suspension at OD_600_ = 10 in the incubation chamber, flanked by 0.4 µm of polycarbonate membranes, and applied a genomic DNA isolation protocol, starting with a two-hour digestion with lysozyme buffer at 37 °C, followed by overnight digestion with lysis buffer at 56 °C. After extensive washing with H_2_O, a highly viscous solution was recovered from the incubation chamber in every instance. Agarose gel electrophoresis showed that the isolated DNA had a high molecular weight and was free of RNA contamination (Figure 2). While the spectral properties were next to optimal or optimal in the case of OD_260_/OD_280_ (>1.7 indicating no contamination with protein), the OD_260_/OD_230_ values were variable depending on the bacterial isolate. The results, demonstrating DNA quality, integrity and yield, are shown in Table 1 and Figure 2. Incubation in the 1 mL incubation chamber and the isolation of DNA from plate-grown cells without lysozyme incubation led to unsatisfactory OD_260_/OD_230_ ratios (Figure 2; Table 2).

In cases of an OD_260_/OD_230_ ratio lower than 1.4, we applied a CHCl_3_ extraction step for DNA purification and subsequently ethanol-precipitated the genomic DNA, which substantially improved the OD_260_/OD_230_ ratio in the majority of cases (Figure 2; Table 2).

Agarose gel run under standard conditions with a constant electric field (and variations thereof) does, however, not resolve DNA fragments larger than 40–60 kpb. In order to assess the size distribution of the DNA isolated via the application of the incubation chamber, we applied pulsed-field gel electrophoresis, which enables the separation of DNA fragments from the low kbp up to the Mbp range [13]. The results suggest that the vast majority of the genomic DNA isolated by the incubation chamber has a molecular weight larger than 50 kbps, with DNA regularly condensed at the upper compression zone and remaining in the well (an indication of very large or circular DNA molecules [21]) (Figure 3). Shearing by pipetting and further purifications of the genomic DNA by extraction with CHCl_3_ and subsequent ethanol precipitation lowered the molecular weight (compare lanes 6 and 7 (7 = purified sample), 9 and 10 (10 = vortexed sample) and 12 and 13 (13 = purified sample)). Of note, the genomic DNA isolated only with lysis buffer without lysozyme pretreatment also displayed a high molecular weight, satisfactory DNA quality parameters, and an even higher yield than that with lysozyme treatment. However, we have not subjected genomic DNA isolated only by lysis buffer application to third-generation sequencing yet (see below). Upon genome sequence availability, we verified that the DNA fragmentation was due to the isolation procedure and not the phosphorothioation of DNA, which leads to fragmentation in pulsed-field gel electrophoresis runs (as the genetic information encoding the respective modification systems is absent) [22].

In order to investigate the suitability of the isolated genomic DNA for downstream applications, we subjected two genomic DNA samples that were isolated by digestion in the incubation chamber to third-generation sequencing via Oxford Nanopore Techniques (ONT). In addition, we also subjected one sample that had been subsequently extracted by CHCl_3_ to ONT sequencing. Very long read lengths were obtained with all three genomic DNA samples, with sizes up to 460 kbps (Table 3).

## 4. Discussion

Third-generation sequencing, such as PacBio and Oxford Nanopore Technology sequencing, has enabled the unravelling of the correct architecture of even seemingly simple bacterial genomes and the completion of complex genomes with a high number of repetitive sequences. For example, PacBio sequencing, with a contig size of 10 to 20 kbp, allows the identification and verification of large chromosomal inversions in bacterial genomes, which could previously only be resolved by macrorestriction mapping by pulsed-field gel electrophoresis [23,24]. Microbiome characterization has shifted from 16S RNA sequencing to the profiling of entire chromosomes by third-generation sequencing [25]. Oxford Nanopore Technology sequencing with individual contigs up into the megabasepair range contributes to the accurate decoding of human and other complex genomes [26,27,28]. Furthermore, long-read sequencing can accurately decode the complex human loci involved in adaptive immunity and drug metabolisms, such as *CYP2A6*, *CYP2D6* and *HLA-B*, which remain inaccessible by short-read sequencing [29]. However, the consistent observation of ultra-long (>100 kbp) and whale-long (>1 Mbp) reads presents technological challenges in each step, from the isolation of intact ultra-high-molecular-weight genomic DNA to library preparation, and the application of the library to the sequencing platform.

In this work, we designed an incubation chamber to isolate ultra-high-molecular-weight genomic DNA of high purity suitable for third-generation sequencing using Oxford Nanopore Technology. The DNA isolation procedure is not labor-intensive, has only a few manipulative steps, and is resource-friendly. While enzymes like lysozyme and proteinase K are still required, no RNAse A is added, as the RNA is efficiently degraded by the combination of a highly alkaline pH, high temperature and long incubation time [30]. The buffer composition is simple, with chaotropic chemicals such as guanidine hydrochloride or isopropanol not being required. The incubation chamber itself and the 50 mL tube can be reused unlimited times. We also have not tested yet whether the lysozyme step is indeed required to ensure high-quality third-generation sequencing (although the quality parameters are adequate (Figure 1; Table 1)), and equally whether we can reuse the buffers for the purification of additional samples. Although the procedure takes 48 h to be conducted, the overall processing is not time-consuming.

In addition, the procedure requires neither pipetting, extraction nor centrifugation, as any buffer exchange occurs by diffusion, thus avoiding the shearing of the incubated DNA. The principle of purifying DNA by buffer and enzyme diffusion in an incubation chamber has been exemplified previously by the preparation of megabasepair-size genomic DNA in agarose plugs to be separated by pulsed-field gel electrophoresis [13]. Here, however, the extraction of long DNA strands from the agarose plugs remains a challenge. Another recent miniature solution to avoid excessive DNA shearing is the electrohydrodynamic separation of genomic DNA in a microfluidic chamber [31]. Although not directly comparable, as different bacterial strains have been used, the yield and purity of the genomic DNA is comparable or superior to commercially available approaches to isolate (ultra-)high-molecular-weight genomic DNA, including the state-of-the-art Nanobind kit (PacBio, Menio Park, CA, USA), which reports 260/280 nm ratios of approx. 1.8 and 260/230 ratios of 1.2–1.8 for Gram-positive and Gram-negative bacteria ([32]; Nanobind-PanDNA-kit manual). However, the extension of the volume of the incubation chamber from 200 µL to 1 mL led to suboptimal purity (Table 2). Equally essential is the regular buffer exchange at the membrane through tube tilting. In addition, although a flexible sample size is already an advantage of the genomic DNA isolation procedure using the incubation chamber, a miniaturization of the dimensions of the incubation chamber (and potentially an alternative design with multiple parallel chambers similar to cell culture chambers, possibly designed by a 3D printer) will allow the isolation of genomic DNA from even smaller sample sizes. Eventually, a smaller pore size of the membrane will further restrict the diffusion of small DNA molecules [33]; however, the diffusion of small DNA molecules was not explicitly tested in our experimental set-up.

The isolation of genomic DNA with the incubation chamber has technological, experimental and biological advantages. The protocol for the isolation of genomic DNA using the incubation chamber can be optimized in order to allow the optimal preparation of genomic DNA from different organisms and microbiome samples. In addition to Gram-negative bacteria, the initial experiments showed that the procedure is also suitable to isolate high-quality genomic DNA from Gram-positive bacteria. The replacement of lysozymes with a mixture of alternative cell-wall-degrading enzymes to efficiently degrade the cell wall—in particular the peptidoglycan layer—will allow the optimal lysis of Gram-positive bacteria, including *Mycobacterium tuberculosis*. Equally, the procedure can be optimized to isolate genomic DNA from fungi via the inclusion of zymolase and/or other cell-wall-degrading enzymes. Furthermore, the incubation chamber could be used to isolate genomic DNA from plants and algae, which usually requires several manipulative steps for isolation, as well as the isolation of protoplasts and/or nuclei prior to the isolation of the genomic DNA. In this context, alterations in the pore size of the semi-permeable membrane will allow the selective separation of cells/protoplasts or nuclei from contaminating components. Also, genomic DNA is hypothesized to possess a similar or even higher molecular weight when eluted from silica-based nanodisks, which currently represent the state-of-the-art technology for the isolation of high-molecular-weight DNA.

The next challenging step after the isolation of genomic DNA with the incubation chamber is the preservation of the large size of the DNA molecules during library preparation, before loading into the flow cell. To achieve this, slow pipetting procedures and automated slow pipetting have been developed [34]. We envisage that incubation chamber-like designs can also aid in avoiding the extensive shearing of DNA at this step.

## 5. Conclusions

We have developed a procedure for the isolation of genomic DNA with an incubation chamber, and we propose that alterations in the experimental protocol and/or methodology will significantly facilitate the isolation of ultra-high-molecular-weight genomic DNA for ultra-long third-generation sequencing. The system could be particularly suitable for challenging samples in which input is limited but high-quality genomic data are required, such as in immunology and oncology, where the genomic profiling of lowly abundant cell populations is required, as well as in forensic science. In addition, the incubation chamber can potentially be used in alternative applications for the isolation of genomic DNA due to the variable pore size of the barrier membranes [33,34], which allow the diffusion of molecules and components of different sizes across the chamber borders.

## Figures and Tables

**Figure 1 microorganisms-13-00534-f001:**
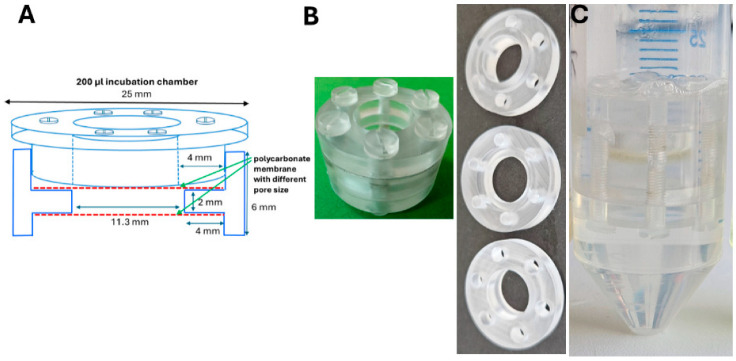
Incubation chamber design. (**A**) Draft of the dimensions of the different parts of the 200 µL incubation chamber. (**B**) Picture of the 200 µL incubation chamber; assembled and individual parts. (**C**) Incubation chamber in 50 mL Eppendorf centrifugation tube, incubating in 10 mL buffer.

**Figure 2 microorganisms-13-00534-f002:**
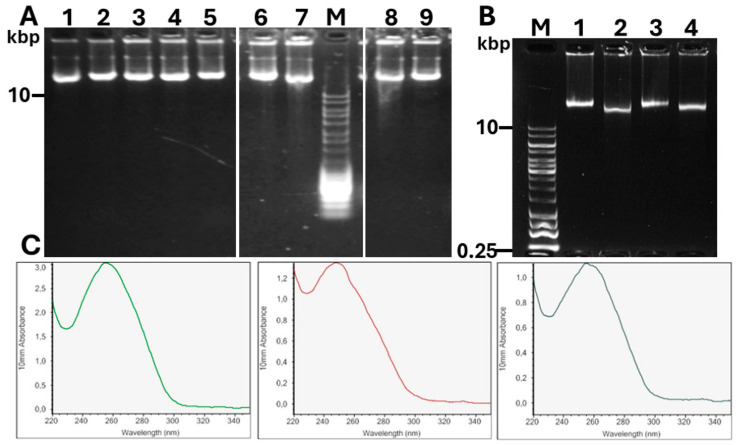
Assessment of quality parameters of genomic DNA isolated by the incubation chamber. (**A**) Separation of genomic DNA isolated by the incubation chamber with 0.8% agarose gel electrophoresis in 1× TAE buffer. DNA samples from (1) *Staphylococcus* sp. W15; (2) *Pseudomonas* sp. AmpSc; (3) *Pseudoarthrobacter* sp. M63; (4) *Arthrobacter* sp. CM63; (5) *Pseudomonas* sp. AmpSc *; (6) *Agrobacterium* sp. AM small **; (7) *Agrobacterium* sp. AM small vortexed **; (8) *Pseudarthrobacter* sp. M63 **; (9) *E. coli* Fec101; (10) *E. coli* Fec101 purified; (M) MassRuler (ThermoFisher); (11) *K. pneumoniae* GT1 mucoid; and (12) *K. pneumoniae* GT1 pdar28. *, 1 mL incubation chamber; ** plate-grown, without lysozyme buffer incubation. (**B**) Separation of genomic DNA isolated by the incubation chamber before and after chloroform extraction. Samples: M, molecular weight marker; genomic DNA from *Agrobacterium* sp. AM small before (1) and after (2) CHCl_3_ extraction. Genomic DNA from *Agrobacterium* sp. AM large before (3) and after (4) CHCl_3_ extraction. (**C**) Spectra of genomic DNA from *K. pneumoniae* GT1 mucoid and *E. coli* Fec101 before and after CHCl_3_ extraction at 220–350 nm, measured by Nanodrop.

**Figure 3 microorganisms-13-00534-f003:**
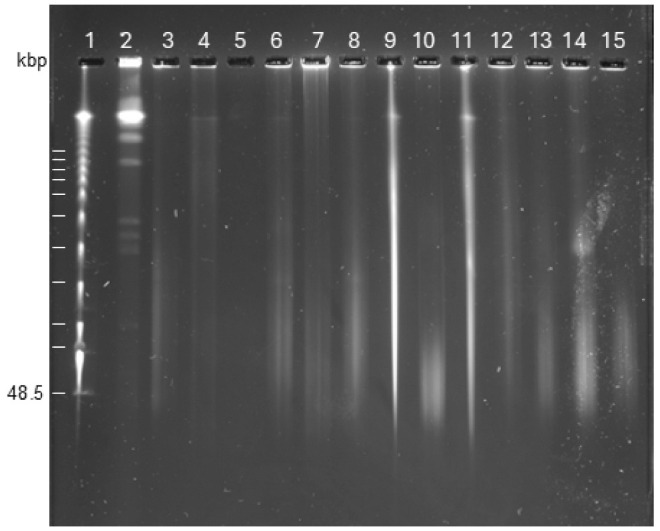
Assessment of size distribution of genomic DNA isolated by the incubation chamber with pulsed-field gel electrophoresis. The genome DNA was separated using a 1.5% agarose gel. Lanes: 1, Lambda DNA; 2, *S. cerevisiae* W303-translocation; 3, *Staphylococcus* sp. W15; 4, *Pseudomonas* sp. AmpSc; 5, *Pseudomonas* sp. AmpSc, purified; 6, *Pseudoarthrobacter* sp. M63; 7, *Pseudoarthrobacter* sp. M63, purified; 8, *Arthrobacter* sp. CM63; 9, *Agrobacterium* sp. AM small *; 10, *Agrobacterium* sp. AM small vortexed *; 11, *Pseudoarthrobacter* sp. M63 *; 12, *E coli* Fec101; 13, *E. coli* Fec101, purified; 14, *K. pneumoniae* mucoid; 15, *K. pneumoniae* pdar28. * plate growth; DNA isolation with only lysis buffer. The Lambda-DNA ladder (lane 1) serves as a molecular weight standard with 48.5 kbp and multiples thereof.

**Table 1 microorganisms-13-00534-t001:** Parameters of DNA quality of bacterial DNA isolated with the incubation chamber.

Organism	Isolation Procedure	OD_600_ of Cell Suspension	DNA Concentration (ng/µL)/Yield [µg]	260/280 Ratio	260/230 Ratio
*Pseudomonas aeruginosa* and *Pseudomonas* sp.					
PRS9	+	10	76.5/15.3	1.87	1.97
GRP3	+	10	161.4/32.3	1.94	1.83
GRP3 *	-	10	196.2/32.5	1.97	1.85
Ear isolate	+	10	135.7/27.1	1.94	1.69
*Klebsiella pneumoniae* GT1					
Mucoid	+	10	90.8/18.2	1.83	1.79
Mucoid **	+	10	153.1/30.6	1.85	1.45
pdar28	+	10	139.2/27.8	1.75	1.93
*Escherichia coli*					
Fec59	+	10	60.5/12.1	1.85	1.51
Fec101	+	10	115.7/23.1	1.88	1.81
Algal microbiome					
*Agrobacterium* AM sp. small	+	10	65.0/13.0	1.93	1.43
*Agrobacterium* AM sp. small *	-	10	232.7/46.5	1.86	1.69
*Agrobacterium* AM sp. small *, **	-	10	286.8/57.4	1.93	1.9
*Agrobacterium* sp. AM large	+	10	78.1/15.6	1.85	1.32
*Agrobacterium* sp. AM large **	+	10	58.7/11.7	1.81	0.72
*Agrobacterium* sp. AM large *	-	10	257.2/51.4	1.9	1.43
Gram-positive bacterium					
*Pseudoarthrobacter* sp. M63	+	10	107.7/21.5	1.72	1.11
*Pseudoarthrobacter* sp. M63 *	-	10	163.3/32.7	2.0	1.71
*Enterococcus muntii* HF	+	10	25.6/5.1	1.99	1.52

* plate-grown, without lysozyme digestion; ** repeated DNA isolation.

**Table 2 microorganisms-13-00534-t002:** OD_260_/OD_280_ and OD_260_/OD_230_ values before and after extraction with CHCl_3_.

Bacteria	Before CHCl_3_ Extraction		After CHCl_3_ Extraction	
Strain	260/280 Ratio	260/230 Ratio	260/280 Ratio	260/230 Ratio
*Agrobacterium* sp. AM small	1.93	1.43	1.78	1.83
*Agrobacterium* sp. AM small *	1.87	1.62	1.87	1.98
*Agrobacterium* sp. AM large *	1.81	0.72	2.0	1.57
*Arthrobacter* sp. CM63	1.75	0.88	1.7	1.19
*Pseudorthrobacter* sp. M63	1.72	1.11	1.72	1.12
*Staphylococcus* sp. W15	1.46	0.67	1.44	0.74
*P. aeruginosa* SG17M ^#^	2.03	0.95	1.97	1.96

* repeated DNA isolation; ^#^ DNA isolation in the 1 mL incubation chamber.

**Table 3 microorganisms-13-00534-t003:** Oxford Nanopore Sequencing quality data.

Strain	Number of Raw Reads	Read Length *N*_50_ [bp]	Total Bases	Longest Read [bp]	Mean Read Quality *
*K. pneumoniae* GT1 mucoid	99,436	8411	539,343,445	459,946	10.3
*K. pneumoniae* GT1 pdar28	247,231	6699	1,177,093,68	168,999	10.1
*Agrobacterium* sp. AM small **	70,978	10,437	471,257,213	143,095	10.4

* Phred quality score calculated as −10 × log (Pe) with Pe estimated probability of error. ** CHCl_3_ extraction.

## Data Availability

The original contributions presented in this study are included in the article. Further inquiries can be directed to the corresponding author.

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
