# Peer review of "Improved Isolation of Ultra-High-Molecular-Weight Genomic DNA Suitable for Third-Generation Sequencing"

_microorganisms, 2025, doi:10.3390/microorganisms13030534_

Round 1

Reviewer 1 Report

Comments and Suggestions for Authors

An article by the Ayse Öykü Ova and co-authors entitled "Improved isolation of ultra-high molecular weight genomic DNA suitable for third generation sequencing" is devoted to the description of a new chamber and a technique for DNA isolation for subsequent sequencing. The work presented by the team of authors is interesting. The authors provide examples of isolating DNA of appropriate quality from a number of Gram-positive and Gram-negative bacteria. The presented results make this article generally useful. However, it does not seem to me that the presented method can be easily replicated in other laboratories, since the manuscript contains the results of not only the proposed protocol, but also the creation of the DNA isolation chamber itself, which will complicate the repetition of this work. Nevertheless, MS corresponds to the profile of the journal Microorganisms, but requires some revision.

So, line 108 - check the punctuation marks.

Line 109 - fully italicize the Latin name Micrasterias papillifera

Line 112. For “E. muntii” provide first full name.

There are few questions about the procedure itself.

1.      Considering the duration of the procedure, which reaches more than 2 days, how convenient can the proposed method be considered?

2.      2. Since the authors use a 50-ml centrifuge tube that requires 10 ml of solution at each stage, how much more expensive is the proposed method compared to those used everywhere?

3.      3. What is the minimum size of DNA that can be isolated using the proposed camera? It is known that the complete characterization of the genome involves sequencing of all types of DNA molecules, not only chromosomal, but also plasmid. Will the pore size of the membrane be small enough to retain small plasmid DNA during dialysis?

Table 1. Please explain the difference between Klebsiella pneumoniae GT1 samples on the first and second lines (mucoid - mucoid).

In Table 1, Parameters of DNA quality of bacterial DNA isolated with the incubation chamber, there is no data on the results of DNA isolation from gram-positive staphylococci.

Author Response

An article by the Ayse Öykü Ova and co-authors entitled "Improved isolation of ultra-high molecular weight genomic DNA suitable for third generation sequencing" is devoted to the description of a new chamber and a technique for DNA isolation for subsequent sequencing. The work presented by the team of authors is interesting. The authors provide examples of isolating DNA of appropriate quality from a number of Gram-positive and Gram-negative bacteria. The presented results make this article generally useful. However, it does not seem to me that the presented method can be easily replicated in other laboratories, since the manuscript contains the results of not only the proposed protocol, but also the creation of the DNA isolation chamber itself, which will complicate the repetition of this work. Nevertheless, MS corresponds to the profile of the journal Microorganisms, but requires some revision.

Individually constructed devices not commercially available are reported in the literature. However, we have added a sentence that the authors can be contacted for more information.

So, line 108 - check the punctuation marks.

Done

Line 109 - fully italicize the Latin name Micrasterias papillifera

Done

Line 112. For “E. muntii” provide first full name.

Done

There are few questions about the procedure itself.

  1. Considering the duration of the procedure, which reaches more than 2 days, how convenient can the proposed method be considered?

Although the isolation of genomic DNA takes two days, the time to prepare and perform the actual steps is not very long. We have added a sentence in the Discussion section: Although the procedure takes 48 hours to be conducted, the overall process is not time consuming.

  1. Since the authors use a 50-ml centrifuge tube that requires 10 ml of solution at each stage, how much more expensive is the proposed method compared to those used everywhere?

We have not calculated exactly, but we estimate that the incubation chamber isolation is less costly overall. We argue not with the following statements in the discussion:

While enzymes like lysozyme and proteinase K are still required, no RNAse A is added as the RNA is efficiently degraded by the combination of high alkaline pH, high temperature and long incubation time [28]. The buffer composition is simple with chaotropic chemicals such as guanidine hydrochloride or isopropanol not required. The incubation chamber itsself and the 50 ml tube can be reused unlimited times. We have also not tested yet whether the lysozyme step is indeed required to ensure high quality third generation sequencing (although the quality parameters are adequate (Figure 1; Table 1) equally as whether we can reuse the buffers for the purification of additional samples. Although the procedure takes 48 hours to be conducted, the overall processing is not time consuming.

  1. What is the minimum size of DNA that can be isolated using the proposed camera? It is known that the complete characterization of the genome involves sequencing of all types of DNA molecules, not only chromosomal, but also plasmid. Will the pore size of the membrane be small enough to retain small plasmid DNA during dialysis?

We have not tested this aspect in particular, but the pore size of the membrane can be varied and also still be reduced as already mentioned in the manuscript. We add not in the discussion: however, this fact has not been explicitly tested in our experimental set-up.

Table 1. Please explain the difference between Klebsiella pneumoniae GT1 samples on the first and second lines (mucoid - mucoid).

These are repeats of the same sample in order to demonstrate the reproducibility. This is indicated now in the manuscript.

In Table 1, Parameters of DNA quality of bacterial DNA isolated with the incubation chamber, there is no data on the results of DNA isolation from gram-positive staphylococci.

Staphylococcus sp. W15 (corrected from W14) genomic DNA is shown on the agarose gel in Figure 2, lane 1. We added now the OD ratios before and after CHCl3 extraction of Staphylococcus sp. W15 in Table 2. We already indicate in the Discussion that DNA isolation can be improved, for example, by enzymes that specifically lyse genus-specific such as lysostaphin.

Reviewer 2 Report

Comments and Suggestions for Authors

The article presents an original method that uses a specially made incubation chamber to isolate genomic DNA with ultra-high molecular weight. By minimizing the need for mechanical handling, this chamber reduces the possibility of DNA shearing during cell lysis and DNA separation. Both Gram-positive and Gram-negative bacteria can be used with this procedure, which appears to yield high-quality DNA that works with third-generation sequencing technologies like Nanopore sequencing.

Key data indicate that the chamber's technique consistently isolates UHMW DNA with high yield and purity, and it seems to be less labor-intensive and complex than traditional protocols.

Although the goal of the study (isolating high-quality DNA) is mentioned, it is not explicitly connected to any problems or limitations in the methods used currently.

The difficulties in separating ultra-high molecular weight DNA for third-generation sequencing are not directly related to the historical background.

The design of the incubation chamber is not described in enough detail to be reproducible. For instance, buffer compositions and incubation durations are provided without presenting a clear explanation for their selection, such as the rationale behind the use of a pH of 9.5 or particular temperatures. Claims regarding the incubation chamber design's distinction lack of references to alternative approaches or explanations of advancements.

Specifics such as the precise model of the micromilling machine ("MiniMill GX") are superfluous unless they have a substantial effect on the reproducibility of the procedure.

The reproducibility of strains such as "P. aeruginosa ear isolate" and "Staphylococcus sp. W14" is compromised by their vague descriptions, which lack precise identification and citation to earlier research.

Some samples (such Agrobacterium sp. large*) have extremely low OD260/OD230 ratios, which could indicate contamination; however, neither the results nor the commentary address this.

Although the results show variation in DNA quality (such as OD ratios) throughout bacterial strains, the text does not discuss whether this is because of methodical shortcomings or innate biological variances.

Figure 2 mentions gel electrophoresis, but it offers no quantitative evaluation of DNA integrity. Visual evaluation is a personal experience.

Throughout the text, the phrase "ultra-high molecular weight genomic DNA" is used inconsistently. Molecular weight thresholds (e.g., >1 Mb) should be specified for clarity.

Confidence intervals and standard deviation, which could provide information about variability, are absent from Table 1.

The new incubation chamber method is not sufficiently compared with current procedures, like agarose plug preparation, in the discussion.

Statements like "suitable for third-generation sequencing" are general and lack supporting evidence, such as sequencing performance measures (e.g., read length or error rates).

The incubation chamber's possible drawbacks, such its scalability, cost, or suitability for a variety of sample types, are not covered.

Minor revision

Experiments that belong in the main text are included in the abstract, which is excessively detailed.

The phrase "high molecular weight genomic DNA" is used repeatedly without providing any new detail.

Although fascinating, historical references to the discovery of DNA (such as those involving Friedrich Miescher, Watson, and Crick) may take attention away from the introduction's main point.

It would be more clear if sentences like "The DNA was subsequently run on a 0.8% agarose gel" were more active.

There are no precise significance for terms like "high molecular weight" and "ultra-high molecular weight," which are used interchangeably.

Author Response

Comments and Suggestions for Authors

The article presents an original method that uses a specially made incubation chamber to isolate genomic DNA with ultra-high molecular weight. By minimizing the need for mechanical handling, this chamber reduces the possibility of DNA shearing during cell lysis and DNA separation. Both Gram-positive and Gram-negative bacteria can be used with this procedure, which appears to yield high-quality DNA that works with third-generation sequencing technologies like Nanopore sequencing.

Key data indicate that the chamber's technique consistently isolates UHMW DNA with high yield and purity, and it seems to be less labor-intensive and complex than traditional protocols.

Although the goal of the study (isolating high-quality DNA) is mentioned, it is not explicitly connected to any problems or limitations in the methods used currently.

We mention potential applications and improvements, but this article describes the proof of principle.

The difficulties in separating ultra-high molecular weight DNA for third-generation sequencing are not directly related to the historical background.

We agree, but wanted to give a more general introduction about the impact of DNA and thus the ability to accurately determine its sequence.

The design of the incubation chamber is not described in enough detail to be reproducible. For instance, buffer compositions and incubation durations are provided without presenting a clear explanation for their selection, such as the rationale behind the use of a pH of 9.5 or particular temperatures. Claims regarding the incubation chamber design's distinction lack of references to alternative approaches or explanations of advancements.

We now provide a reference for the hydrolysis of RNA at alkaline pH (Li and Breaker, J Am Chem Soc, 1999) and describe the optimum for catalysis of proteinase K in the Materials and Methods.

We also add

High pH and detergent provide optimal conditions for the catalytic activity of the serine protease proteinase K, with EDTA to inhibit Mg2+-dependent nucleases.

Specifics such as the precise model of the micromilling machine ("MiniMill GX") are superfluous unless they have a substantial effect on the reproducibility of the procedure.

We prefer to keep the precise model.

The reproducibility of strains such as "P. aeruginosa ear isolate" and "Staphylococcus sp. W14" is compromised by their vague descriptions, which lack precise identification and citation to earlier research.

Indeed, those strains have not been reported by us with respect to specific biological features, but might be reported in the future. In any case, as reported in this manuscript, they are available for other researchers upon request as is common practice.

Some samples (such Agrobacterium sp. large*) have extremely low OD260/OD230 ratios, which could indicate contamination; however, neither the results nor the commentary address this.

We agree, but have added this sample as the quality can be readily recovered by CHCl3 extraction (now added in Table 2) and as an example that care must be taken to allow buffer exchange during the incubation time. We add now in the discussion: However, extension of the volume of the incubation chamber from 200 µl to 1 ml led to suboptimal purity. Equally essential is the regular buffer exchange at the membrane through tube tilting.

Although the results show variation in DNA quality (such as OD ratios) throughout bacterial strains, the text does not discuss whether this is because of methodical shortcomings or innate biological variances.

We have now added further discussion on shortcomings (see answer to the previous comment) equally as already present comments with respect to optimization of procedure by choosing alternative enzymes for cell wall digestion etc. Repeats are methodical shortcomings indicating that handling can be improved. We indicate not also additional repeats equally as provide examples of DNA samples isolated without the lysozyme step.

Figure 2 mentions gel electrophoresis, but it offers no quantitative evaluation of DNA integrity. Visual evaluation is a personal experience.

We agree, however, quantitative systems like the TapeStation also do not separate DNA above 60 kbp in length and the length quality of the DNA and the suitability of the genomic DNA isolated by our method for third generation sequencing with long DNA molecules to be sequenced is sufficiently shown. In any case, we acknowledge the criticism of the reviewer and have now analyzed our DNA samples by pulsed-field gel electrophoresis. We include these results as Figure 3. The results show that the vast majority of the DNA fragments are over 50 kbp in size.

Of note, in pulsed-field gel electrophoresis DNA fragmentation can also be observed upon phosphorothioate modifications. However, we have no genetic (Klebsiella pneumoniae, Agrobacterium sp.) nor experimental (Escherichia coli Fec101) indication a phosphorothioation system is present in the investigated strains.

Throughout the text, the phrase "ultra-high molecular weight genomic DNA" is used inconsistently. Molecular weight thresholds (e.g., >1 Mb) should be specified for clarity.

We have revised the text to discriminate between high and ultra-high molecular weight DNA more specifically. The literature discriminates between ultra-long (>100 kbp) and whale long (>1 Mbp) reads (as mentioned in the discussion) and we define now in the introduction ultra-long DNA strands (> 1 Mbp).

Confidence intervals and standard deviation, which could provide information about variability, are absent from Table 1.

We have not repeated each sample more than two times. This is also not standard for DNA isolation and sequencing procedures. Thus, we cannot do reliable statistics. However, we exemplarily report quality values for biologically independent isolation of DNA in Table 1 and added the quality parameters of additional genomic DNA samples including samples without lysozyme buffer treatment.

The new incubation chamber method is not sufficiently compared with current procedures, like agarose plug preparation, in the discussion.

We have now compared the incubation chamber method in more detail with the agarose plug preparation and with a very recently described ultra-high molecular weight DNA isolation approach at low shear forces based on electrohydrodynamic separation (PMID: 39847326) that came to our knowledge after the submission of this manuscript. We write now:

The principle of purifying DNA of buffer and enzyme diffusion similar as in the incubation chamber has been exemplified previously by the preparation of Mega-base pair size genomic DNA in agarose plugs to be separated by pulsed-field gel electrophoresis [13]. Here, however, the extraction of the long DNA strands from the agarose plugs remains a challenge. Another recent solution to avoid excessive DNA shearing is the electrohydrodynamic separation of genomic DNA in a microfluidic chamber [29].

Statements like "suitable for third-generation sequencing" are general and lack supporting evidence, such as sequencing performance measures (e.g., read length or error rates).

We have provided read length and error rate (mean read quality) in Table 3. We have however now provided the commonly used definition for error rate calculation in Table 3. We have also uploaded the raw sequencing reads to NCBI under BioProject ID submission PRJNA1219145. This information has been added to the manuscript.

The incubation chamber's possible drawbacks, such its scalability, cost, or suitability for a variety of sample types, are not covered.

We have added (and already present) in the discussion information about cost estimation (chemicals/enzymes required/not required, unlimited reusability of the chamber, potential buffer reuse). We believe that the incubation chamber has the potential to be used for a variety of sample types which we have also already discussed.

Minor revision

Experiments that belong in the main text are included in the abstract, which is excessively detailed.

Abstract has been revised.

The phrase "high molecular weight genomic DNA" is used repeatedly without providing any new detail.

We have now run a pulsed-field gel electrophoresis to estimate the DNA size.

Although fascinating, historical references to the discovery of DNA (such as those involving Friedrich Miescher, Watson, and Crick) may take attention away from the introduction's main point.

We wanted to place the impact of DNA into a larger context.

It would be more clear if sentences like "The DNA was subsequently run on a 0.8% agarose gel" were more active.

Revised to specify genomic DNA.

There are no precise significance for terms like "high molecular weight" and "ultra-high molecular weight," which are used interchangeably.

We have revised the text to discriminate between high and ultra-high molecular weight DNA more specifically.

In addition to the reviewers comments, we have more clearly and accurately defined the DNA binding principle in the discussion.

For additional comparison of the procedure, according to the reviewers’ questions, we have now also added quality parameters for DNA isolated without the lysozyme buffer step prior to cell lysis in Table 1.

Round 2

Reviewer 2 Report

Comments and Suggestions for Authors

The article has been revised in many parts, clarifying concerns about the new method and making the necessary changes based on the suggestions provided. No further modifications are required.